# OpenReview forum: "MOSAIC: Modular Foundation Models for Assistive and Interactive Cooking"
_robot-learning.org/CoRL/2024/Conference — CoRL 2024_

### Official Review · Reviewer_ibvm · 2024-07-16
**Interesting robot systems for (multiple) human robot interaction for everyday kitchen tasks**

**Originality:** 3
**Technical Quality:** 4
**Clarity Of Presentation:** 4
**Potential Impact:** 2
**Recommendation:** 3
**Confidence:** 4

**Review:**

The paper has a few noteworthy strengths. First of all, the authors attempt to tackle some of the most ambitious robot-human collaboration tasks in the real world, with natural language interface and open vocabulary. The proposed approach seems like a well-designed, modular system that pieces together several SOTA models across AI fields (e.g. object detection, LLM task planning, etc). The combination of behavior trees with LLMs for task planning is fairly interesting. The most impressive aspect of the paper is that it includes comprehensive end-to-end experimental evaluation across 6 different long-horizon tasks, as well as per module evaluation with ablation studies and detailed analysis. It’s appreciated that each of the error types from the robots are carefully broken down, repeated, analyzed and remedied as much as possible.

However, the paper is not without room for improvement. First of all, one may be concerned about the generality of the approach (e.g. what if the recipe is unknown, what if no map is given for navigation, what if some skills are difficult to learn with RL or motion planning). Honestly, a final success rate of 68% is great, but are we really 68% there in terms of a helpful robot assistant in our kitchen? In the limitation section, it’s mentioned that “expanding to new environments in a scalable and flexible manner may require us to revisit previous assumptions and adopt new capabilities”. It would be helpful if the authors can explicitly list out all the (hidden) assumptions of the proposed method, i.e. what breaks if the method is deployed in a brand new kitchen with a brand new user companion. Lastly, it’s a bit unclear whether this paper suits the CoRL venue because to the best of my knowledge, the only components trained by the authors are the pick skill policy and the human forcaster, whereas all other components are off-the-shelf pretrained models or engineered solutions (e.g. LLMs, OWL object detector, motion planners, etc)

**Quality Of The Limitations Section:**

3

**Questions For Rebuttal:**

- How general is the proposed method? What are all the assumptions that the method has?
- How does the task planner break down the overall task into subtasks? Is it via LLM zero-shot capabilities, or are the recipes provided to the system?
- Pick skill still seems like a major bottleneck for the overall system. Any idea on how to further improve it?

**Robotics Focus:**

4

**Summary Of Paper:**

The paper presents MOSAIC, a modular architecture for coordinating multiple robots to interact with users using natural language, and manipulate an open vocabulary of everyday objects to solve complex cooking tasks. The main contributions of this paper are: 1) a modular cooking assistant that use LLMs for interactive task planning, VLMs for visuomotor skills, and motion forecasting models to predict human intents for collaboration, 2) perform comprehensive evaluation of MOSAIC, either end-to-end, or per-module, 3) draw insights and lessons learned.

**Summary Of Recommendation:**

I would recommend weak accept since despite the room of improvement mentioned above, I believe the community still benefits a lot from reading and discussing this work due to its thorough experimental evaluation.

---

### Official Review · Reviewer_34B7 · 2024-07-19
**MOSAIC Review**

**Originality:** 3
**Technical Quality:** 3
**Clarity Of Presentation:** 4
**Potential Impact:** 3
**Recommendation:** 3
**Confidence:** 4

**Review:**

MOSAIC builds an impressive system for interactive situated cooking with humans. Human users are able to interact with MOSAIC through natural language with the LLM-based task planner, as well as physically with the human motion forecasting model. This setting is particularly challenging due to the safety concerns when interacting closely with humans, which MOSAIC addresses with a trained motion forecasting model that stops the robot when a human is approaching a close proximity. This is a large systems effort, with many real robot evaluations validating the efficacy of each individual component. The paper was well written and the supplemental video was appreciated to visualize the interactive rollout of MOSAIC.

While MOSAIC provides some impressive demonstration rollouts with expert author users, a larger user study with non-expert users -- users who were not involved with building MOSAIC -- would be needed to validate the assistive claim of the system. Is MOSAIC able to be assistive for a more general audience with different requirements? Are novice users able to learn to use the system effectively?

**Quality Of The Limitations Section:**

3

**Questions For Rebuttal:**

- From Table 1, why were different skills implemented so differently (learned with RL, hand-engineered, and planning-based)? This requires access to more information about the environment, such as a map for planning or access to a simulated environment for the RL training.

- Were there any experiments evaluating the ability to add new tasks/”recipes” to the system? The reviewer is concerned about generalizability of the system in an interactive setting.

- Most critically, how were the interactive and assistive qualities of the system evaluated? Was a user study conducted with the full system? A N>5 user study with users not involved in building MOSAIC would greatly improve support for the claim of interactivity and assistance. Did users feel the robot was helpful? Would users want to use this system again for cooking?

**Robotics Focus:**

4

**Summary Of Paper:**

This paper introduces MOSAIC, an interactive, modular system for assistive cooking. MOSAIC utilizes 3 components: a LLM and DAG-based task planner, a human motion forecasting module, and motion primitives. Using this hierarchical architecture, MOSAIC is able to interact with humans for high-level task planning with natural language while also interacting physically for tasks such as hand-over. Results show MOSAIC is capable of cooking six predefined recipes with a human over 60 trials, as well as a clear evaluation of each module’s performance and contributions to the overall system.

**Summary Of Recommendation:**

I chose my recommendation because the system demonstrates impressive capabilities with expert users, but leaves the question of whether the system is able to adapt to a novice user unanswered.

---

### Official Review · Reviewer_MKTe · 2024-07-22

**Originality:** 3
**Technical Quality:** 2
**Clarity Of Presentation:** 3
**Potential Impact:** 3
**Recommendation:** 3
**Confidence:** 5

**Review:**

Strength:
 - The paper focuses on an important problem, which is human-robot coordination. The topic is also of interest to CoRL attendees.

Weakness:
 - The paper claims that modularization is its main technical contribution. However, several prior studies have already addressed modularized systems and methods to tackle complex, long-horizon tasks. For instance, Voxposer first constructs affordance maps with different LLM and then generates codes. This makes the paper’s technical contribution in this area less significant. However, I do agree that using LLM systems for human-robot interaction is novel.
 - The interactive task planner, which aims to allocate tasks to two robots and humans in the experiment, lacks significant details. It is unclear how the subtasks are allocated, whether the LLM first generates the overall plan, and how frequently the plan is regenerated.
 - The experiment setting is unclear. It will be better to use one running example to show how the method works. And it will be better to add more descriptions for each task.

**Quality Of The Limitations Section:**

3

**Questions For Rebuttal:**

- What are the principles used to design the modularization architecture? Considering the problem from a hierarchical perspective, the task planner handles high-level planning while visuomotor skills focus on the low-level aspects. How should the human motion forecasting module be positioned within this hierarchy? Additionally, there are sensor-specific modules like object detection. Are these sensor-wise modules included in the authors’ modularization framework?
 - Regarding the interactive planner, how are subtasks assigned to different robots and humans?
 - Does the language model first generate the sequence of actions? Are the subtasks shown in the upper right corner of Figure 3 predefined?
 - Do the robots need to obtain permission from humans first before executing these actions?

**Robotics Focus:**

4

**Summary Of Paper:**

The paper proposes a modular system, MOSAIC, to coordinate multiple robots and help humans finish multiple kitchen tasks. It mainly has three components: an interactive task planner, human motion forecasting, and visuomotor skills. Through experiments, it show that the proposed system performs better than a baseline method which only uses one single forward query of the LLM.

**Summary Of Recommendation:**

The paper focuses on using foundation model systems for human-robot collaboration, which is novel. However, the paper’s technical contribution is overclaimed, and the methodology section needs more clarification regarding the design principles and significant details.

---

### Author Rebuttal · Authors · 2024-08-07

We upload the exact prompts used in our task planner, user study, and baseline comparisons in the Appendix of the main paper. We have highlighted all changes from the previous version in blue.

---

### Decision · Program_Chairs · 2024-09-04

**Decision:**

Accept

**Comment:**

Strengths:

- Interesting problem of multi-robot and human coordination, using LLMs for natural language interaction
- Modular architecture, integrating many different SoTA models from literature from object detection, segmentation, human action detection. Impressive overall system, with large number of real-world evaluations.

Weaknesses:

- The interactive talk planner lacks details, including exact prompts used. Please provide details either in the main paper or appendix, as requested by reviewers. Also a discussion on how the planner can be adapted to related but unseen tasks, like a different recipe, or unseen environment, would be helpful.
- While the system is impressive, it seems a little specific to the problem at hand. A better motivation of the system design would help future researchers modify it to other problems. Additionally, please clearly list out the assumptions in the system.
- The real-world experiments are conducted with a small set of expert users, which raises questions about how well the approach generalizes to other users. Including experiments with novice users would make the claims about interactivity and assistance stronger.

Post rebuttal:
Authors have clarified the assumptions and details of the approach. They have also provided a description of the approach's ability to generalize to new tasks, and presented a larger user study with novice users. Overall this is a good paper.